# Locally adaptive temperature response of vegetative growth in *Arabidopsis thaliana*

**Pieter Clauw[1]\*, Envel Kerdaffrec[2], Joanna Gunis[1], Ilka Reichardt-Gomez[3], Viktoria Nizhynska[1], Stefanie Koemeda[4], Jakub Jez[4], Magnus Nordborg[1]\***

[1]Gregor Mendel Institute of Molecular Plant Biology, Austrian Academy of Sciences, Vienna BioCenter, Vienna, Austria; [2]Department of Biology, University of Fribourg, Fribourg, Switzerland; [3]Max Planck Institute of Molecular Cell Biology and Genetics, Dresden, Germany; [4]Plant Sciences Facility, Vienna BioCenter Core Facilities GmbH, Vienna, Austria

**Abstract** We investigated early vegetative growth of natural *Arabidopsis thaliana* accessions in cold, nonfreezing temperatures, similar to temperatures these plants naturally encounter in fall at northern latitudes. We found that accessions from northern latitudes produced larger seedlings than accessions from southern latitudes, partly as a result of larger seed size. However, their subsequent vegetative growth when exposed to colder temperatures was slower. The difference was too large to be explained by random population differentiation, and is thus suggestive of local adaptation, a notion that is further supported by substantial transcriptome and metabolome changes in northern accessions. We hypothesize that the reduced growth of northern accessions is an adaptive response and a consequence of reallocating resources toward cold acclimation and winter survival.

## Editor's evaluation

The article combines genetic and phenotypic approaches to show convincing evidence of local adaptation in early vegetative growth of *Arabidopsis* lineages sampled from a wide range of locations. The authors show larger initial size and slower growth of northern accessions compared to southern accessions when exposed to cold temperatures, suggesting that northern accessions potentially reallocate resources for winter survival. This study is commendable for its scope and comprehensive analysis of local adaptation of a highly polygenic trait in a model weed.

**\*For correspondence:**
pieter.clauw@gmi.oeaw.ac.at (PC);
magnus.nordborg@gmi.oeaw.ac.at (MN)

## Introduction

Plants use a wide variety of life history strategies in adaptation to their local environment. These strategies have evolved to maximize fitness, but are constrained by trade-offs between components such as growth, survival, and reproduction (*Lande, 1982*; *Stearns, 1992*). While most life history studies investigate differences between species, there is also variation found within species, including in *Arabidopsis thaliana*, where life history variation has been linked to climate parameters (*Estarague et al., 2022*; *Vasseur et al., 2018*; *Sartori et al., 2019*). Less clear is how trade-offs are shaping this variation. In this article, we consider vegetative growth, a key component of life history, and use transcriptome and metabolome data to help explore potential trade-offs. We did this specifically in cold temperatures meant to simulate natural conditions in the northern regions of the species distribution.

Local adaptation studies in *A. thaliana* have found important roles for life history traits such as seed dormancy and flowering time (*Takou et al., 2019*). Temperature is a major regulator of these traits and local populations are adapted to their local climate (*Martínez-Berdeja et al., 2020*; *Simpson and Dean, 2002*; *Hepworth et al., 2018*). Plant growth is also affected by temperature, and previous

studies have detected genetic variation underlying growth-related traits (*Bac-Molenaar et al., 2015*; *Marchadier et al., 2019*), as well as signals of polygenic adaptation (*Wieters et al., 2021*). How vegetative growth is adapted to local temperatures remains unclear, however.

Growth can be seen as the end sum of a vast number of physiological processes. All of these are genetically determined but can also be heavily influenced by environmental conditions (*Bac-Molenaar et al., 2015*; *Fritz et al., 2018*). Growth is therefore not only genetically a complex trait but also enormously plastic. The most straightforward environmental effect is when conditions are so adverse that growth reaches a physiological limit, making it impossible for the plant to grow any further. This is called 'passive plasticity' (*Forsman, 2015*; *van Kleunen and Fischer, 2005*). Yet, when survival is at stake, it may also be in the interest of the plant to actively inhibit growth upon deteriorating environmental conditions (*Claeys and Inzé, 2013*), called 'active plasticity' (*Forsman, 2015*; *van Kleunen and Fischer, 2005*). Since vegetative growth ultimately determines photosynthetic surface and thus energy input that can be invested in the next generation, it is directly related to fitness, and is typically in trade-off with survival. Allocation of resources towards either growth or survival is thus an important balance to keep, and plants are expected to be adapted to constantly perceiving and responding to specific environmental changes as cues for looming adverse conditions.

Cold acclimation is a well-studied mechanism in plants, in which decreasing temperatures induce freezing tolerance in preparation for winter (*Thomashow, 1999*; *Hughes and Dunn, 1996*). This

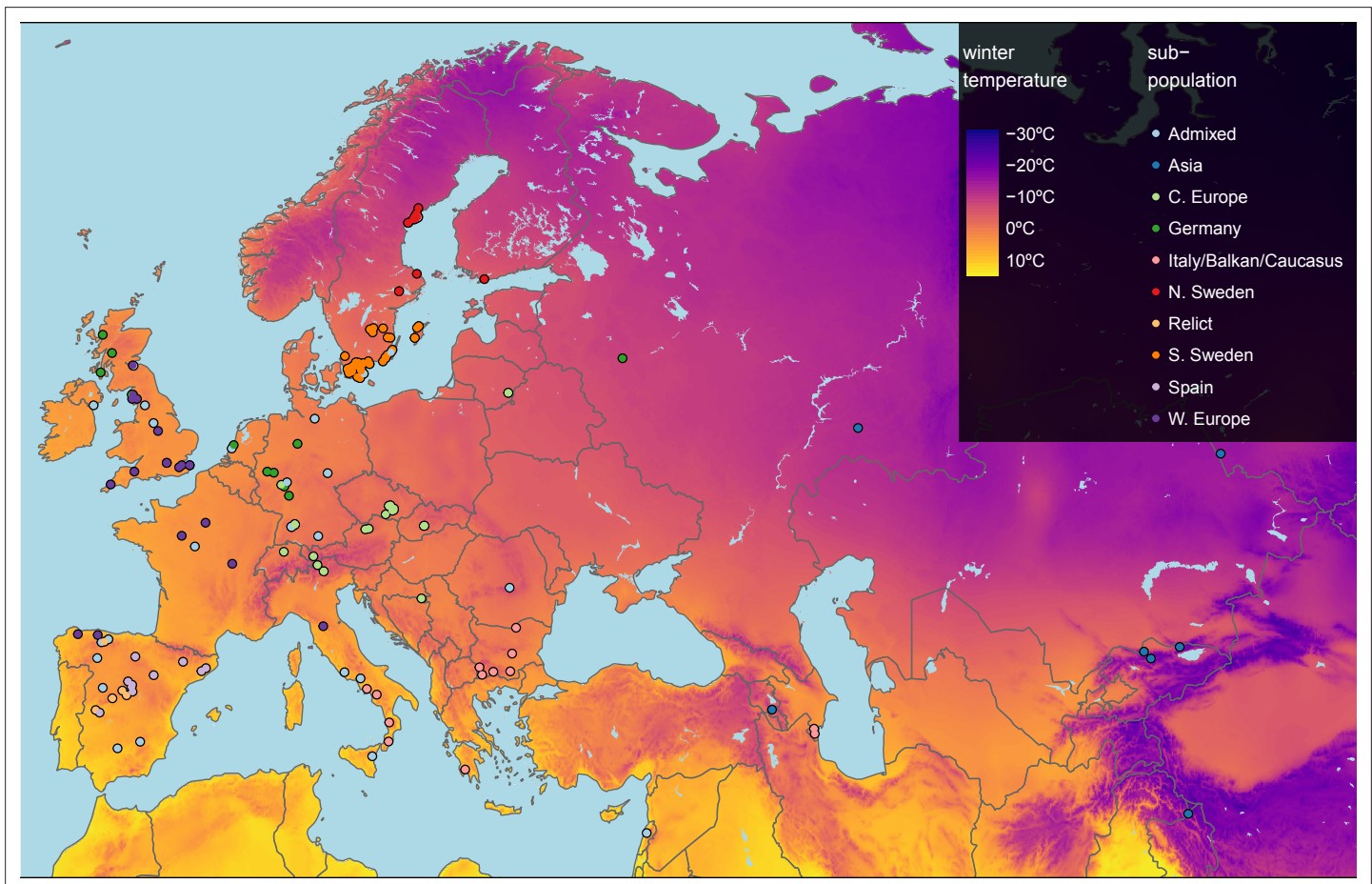

**Figure 1.** Geographic origin of the 249 accessions. Map color shows winter temperature (mean temperature of coldest quarter). Accessions are colored according to subpopulation (*1001 Genomes Consortium, 2016*). Accessions from the warmest and coldest regions are from Greece and the Himalayas, respectively.

The online version of this article includes the following source data and figure supplement(s) for figure 1:

**Figure supplement 1.** Timeline of the experiments.

**Source data 1.** List of all 249 accessions with indication of the 8 accessions used for the transcriptome analysis.

temperature response is typically studied at 4°C, but has been observed in temperatures up to 12°C (**Bond et al., 2011**). The increased freezing tolerance is accomplished by changing membrane composition, producing cryoprotective polypeptides such as COR15A (**Artus et al., 1996**; **Steponkus et al., 1998**) and accumulating compatible solutes with cryoprotective properties such as raffinose, sucrose, and proline (**Nanjo et al., 1999**; **Gilmour et al., 2000**; **Taji et al., 2002**). Main regulators of cold acclimation are CBF1/DREB1b, CBF2/DREB1c, and CBF3/DREB1a, three AP2/ERF transcription factors, for which allelic variation in CBF2 has been linked to natural variation in freezing tolerance (**Oakley et al., 2014**; **Park et al., 2018**; **Alonso-Blanco et al., 2005**). Nothing is known, however, about whether natural variation in freezing tolerance regulation also influences a trade-off with growth responses to cold temperatures.

Here, we investigated the role of growth in adaptation to cold temperatures by comparing vegetative growth of 249 accessions (**Figure 1**) grown in daily maximum temperatures of 16 and 6°C for a period of 3 weeks following seedling establishment (**Figure 1—figure supplement 1**). Rosette growth of each plant was measured twice a day during temperature treatments using automated phenotyping. The experiment generated rosette growth estimates at a high temporal resolution in two ecologically realistic temperature conditions in a wide set of accessions, allowing us to look for patterns of local adaptation.

## Results

### Estimating plant growth

Our highly replicated experiment yielded dense (two measurements per day) time-series growth data for over 7000 individual plants (5 replicate plants × 249 accessions × 2 treatments × 3 replicate experiments). These data were used to model plant growth and estimate growth parameters for further analysis. Unlimited growth should be exponential, but plant growth is known to slow down with increasing size, and therefore a power-law function, $\frac{dM}{dt} = rM^{\beta}$, with $\beta < 1$ is typically a better fit than a pure exponential function (for which $\beta = 1$ — in the equation, $M$ is the size, $r$ is the growth rate, and $\beta$ is a scaling factor that allows rate of size increase to change with size). Growth according to a power-law function typically describes early stages of plant growth especially well (**Paine et al., 2012**), and our rosette size measurements were no exception (see 'Materials and methods'). To calculate the rosette size from a power-law function at a given time point, only three parameters are required: the initial size ($M_0$), growth rate ($r$), and $\beta$. Note that ($M_0$) is the rosette size at the start of the temperature treatment 14 days after stratification (**Figure 1—figure supplement 1**) and is thus not affected by the temperature treatment. We used a nonlinear mixed model to obtain estimates for the initial size, growth rates, and $\beta$. Accession was added as fixed effect for initial size and growth rate, temperature and accession × temperature interactions were added as fixed effects for growth rate only. $\beta$ was considered to be constant over accessions and temperatures. The 'temperature response' of the growth rate was calculated for each accession as the slope between the growth rate at 16 and 6°C. As expected, all accessions grew faster when it was warmer. The observed phenotypic variation (**Figure 2—figure supplement 1**) is to a large extent explained by genetic variation; broad-sense heritabilities are 0.41 for initial size, and 0.57 and 0.32 for growth rate at 16 and 6°C, respectively.

### Growth parameters correlate with the environment of origin

If growth rates are locally adaptive, they may reflect the environment of origin of each accession. To investigate this, we correlated our estimated growth rates with climate data. The climate variables showing the strongest correlations with the different growth parameters were linked to winter temperatures (**Figure 2—figure supplement 2**), also when correlations were corrected for population structure (**Figure 2—figure supplement 2**). In particular, the mean temperature during the coldest quarter (henceforth referred to as 'winter temperature') was most strongly correlated with our parameter estimates, and we focus on it in what follows.

#### Initial size

Accessions from colder climates generally had higher initial rosette size ($M_0$), 2 weeks after germination, than accessions from warmer climates ($r = -0.39$), but then grew more slowly during the temperature experiment – regardless of temperature regime (**Figure 2**, **Figure 2—figure supplement**

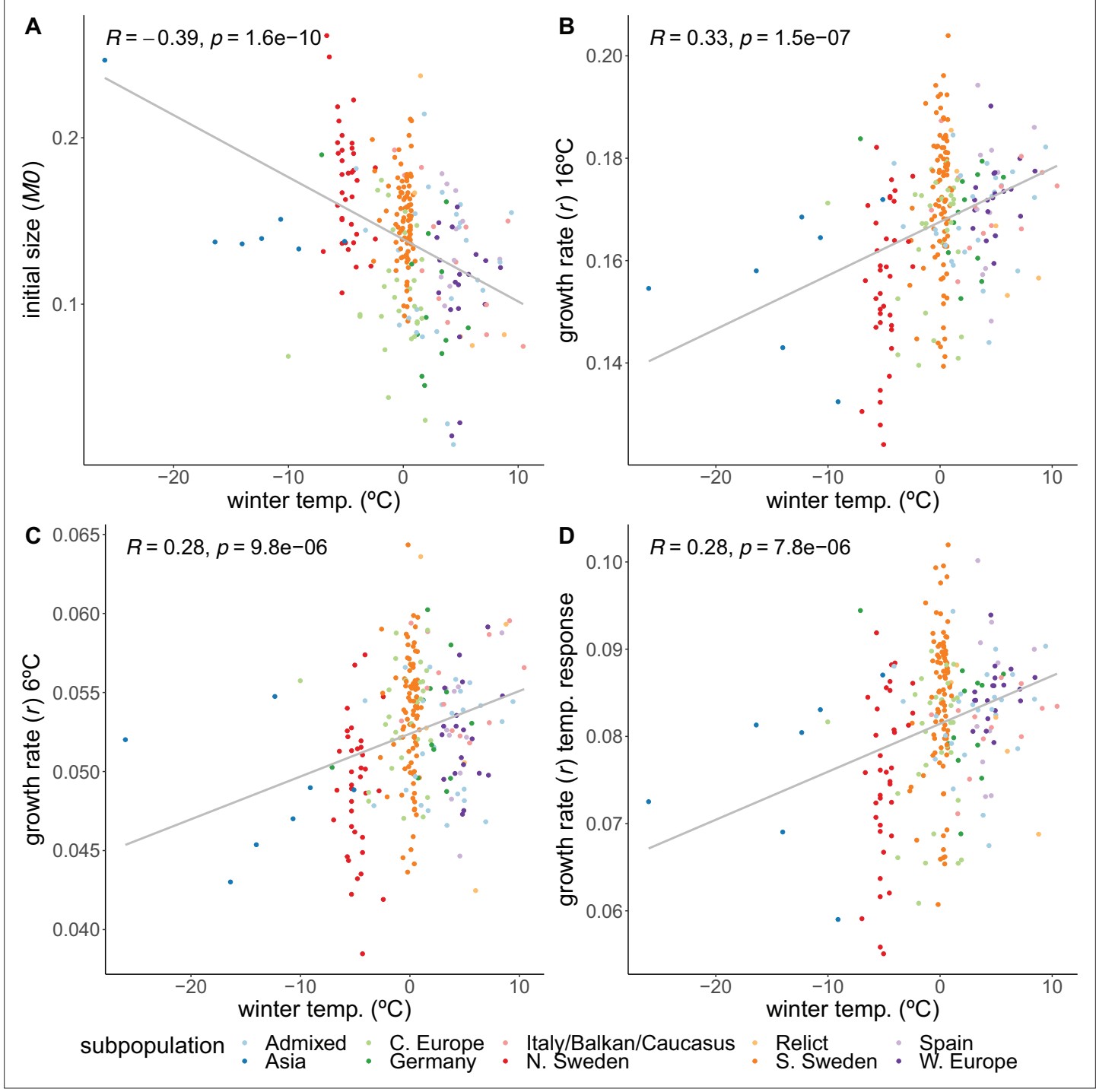

**Figure 2.** Correlations of growth parameters with winter temperature. (**A**) Initial size. (**B**) Growth rate at 16°C. (**C**) Growth rate at 6°C. (**D**) Temperature response of growth rate. Colors indicate genetically defined subpopulations of the accessions (*1001 Genomes Consortium, 2016*).

The online version of this article includes the following figure supplement(s) for figure 2:

**Figure supplement 1.** Variation among accessions of initial size ($M_0$), growth rate ($r$), and the temperature response of the growth rate.

**Figure supplement 2.** Correlations between growth parameters and (bio)climate variables.

**Figure supplement 3.** Population structure-corrected correlations between growth parameters and (bio)climate variables.

**Figure supplement 4.** Correlations of growth parameters with winter temperature, excluding accessions defined as Asian subpopulation.

**Figure supplement 5.** Seed size correlations.

**Figure supplement 6.** Growth rate's temperature response variation.

*3*). Because during the first 2 weeks the plants were growing at 21°C, it is impossible to disentangle early growth from growth at a warmer temperature.

One reason for this pattern is likely to be the differences in seed size between accessions. Using unpublished seed size measurements for a subset of 123 Swedish accessions from previous experiments, we found that seed size is positively correlated with initial size ($r = 0.28$; *Figure 2—figure supplement 5A*), and also with winter temperature ($r = -0.75$; *Figure 2—figure supplement 5B*), at least for the subset of 123 Swedish accessions. In a random-effect model, winter temperature explained 32.7% of the variation in initial size, whereas seed size explained 11.9%. Winter temperature is still significantly associated with initial size when seed size is taken into account ($p-value < 1e - 04$). The precise role of seed size in initial growth is surely a topic that would benefit from further studies, but for the purpose of this study, it is clear that seed size alone cannot explain the geographic pattern we observe for $M_0$ and that there must be a role for variation in growth rate during the very initial phases of seedling growth.

## Growth rates

While the initial sizes correlate negatively with winter temperature, we observed the opposite relation for the growth rates. Despite being larger initially, accessions from colder climates grew more slowly during both the 16°C ($r = 0.33$) and 6°C ($r = 0.28$) treatments (*Figure 2*). The higher growth rates of accessions from warmer climates prove that resources are not limiting, suggesting that the northern lines are actively inhibiting their growth, and that growing slower may be beneficial in colder climates, perhaps in preparation for winter. Accessions from colder climates were also less sensitive to the temperature experiment in the sense that the temperature response of the growth rate increased with winter temperature of origin ($r = 0.28$; *Figure 2D*). Even though accessions from the Asian and north Swedish subpopulations were more variable in their growth rate temperature response (*Figure 2—figure supplement 4A*), the correlations still hold when removing either Asian or northern and southern Swedish subpopulations (*Figure 2—figure supplement 6B*, *Figure 2—figure supplement 4D*), and when looking specifically within the northern and southern Swedish subpopulations (*Figure 2—figure supplement 6*).

## Cold acclimation response

Just like the observed geographic pattern of the growth rates, metabolite measurements taken at the final day of our experiment and presented in an earlier publication (*Weiszmann et al., 2020*) showed clear differences between accessions from cold and warm regions, and many of these differences involved metabolites with a known role in cold acclimation. Since the transcriptomic component of cold acclimation is well studied, we analyzed the expression profiles of 251 previously described cold-acclimation genes (*Figure 3—source data 1*) in eight accessions that were representative in terms of their growth and metabolome profiles (*Figure 3—figure supplement 1*). The selected genes are described in the literature as being differentially expressed upon exposure to cold, and their expression is under control of at least one of the known transcription factors regulating cold acclimation: CBF1, CBF2, CBF3, HSFC1 (*Park et al., 2015*), or ZAT12 (*Vogel et al., 2005*). In our experiment, expression of these genes is likewise more affected by temperature than expected by chance (*Figure 3*; $\chi^2$-test: $p - value < 0.001$) and separates the two accessions from the coldest region (northern Sweden) from the rest in the 16°C treatment, and the three accessions from the warmest regions (Spain and Azerbaijan) from the rest in the 6°C treatment. Expression of different subsets of the selected cold-acclimation genes shows clear correlations with winter temperature of origin (*Figure 3—figure supplement 2*, *Figure 3—figure supplement 3*). In particular, the genes that were previously found to be upregulated upon cold exposure showed higher expression in accessions from cold climates (*Figure 3—figure supplement 4*). Since the expression of these cold-acclimation genes has been linked to the strength of cold acclimation in previous experiments (*Park et al., 2015*; *Vogel et al., 2005*), these accessions likely differ in their ability to cope with freezing temperatures upon cold treatment.

## Growth is polygenic and shows signs of local adaptation

We used genome-wide association to investigate the genetic architecture underlying variation for the different growth parameters. As expected, these traits appear to be highly polygenic, and there

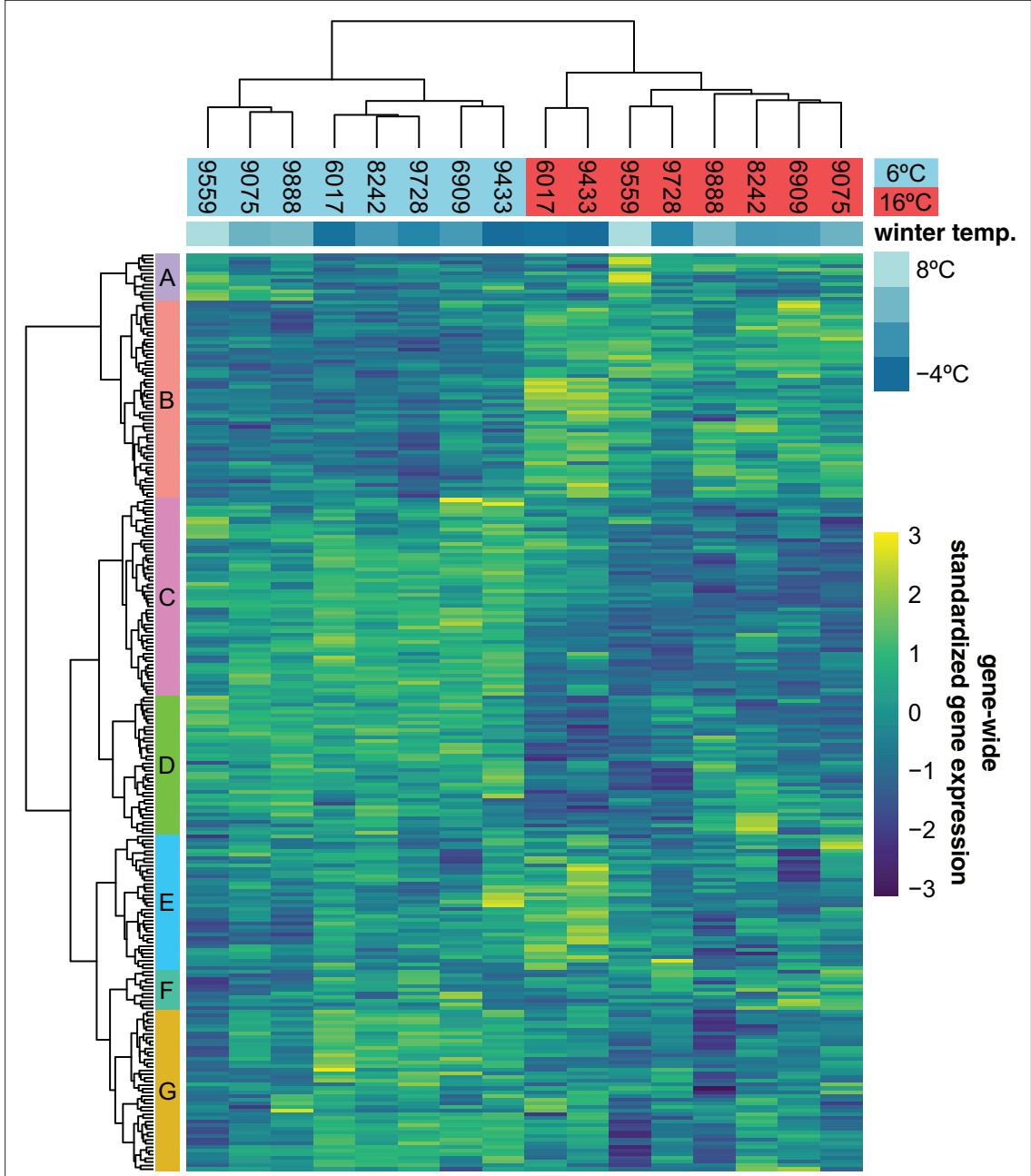

**Figure 3.** Expression of 251 previously described cold-acclimation genes. Expression is shown as the gene-wide z-scores of the normalized counts. The z-scores allow for grouping genes with a similar expression behavior over the different accessions in both temperatures. The top bar indicates winter temperature (°C) for each accession's origin. Both dendrograms along y-axis and x-axis, respectively, show hierarchical clustering of genes, and of accessions in both temperatures.

The online version of this article includes the following source data and figure supplement(s) for figure 3:

**Figure supplement 1.** Growth parameters and metabolic distance of RNA-sequenced accessions in relation to local mean temperature of coldest quarter.

**Figure supplement 2.** Cluster-specific expression in relation to winter temperature.

**Figure supplement 3.** Proportion of genes in each cluster for which expression significantly correlated with winter temperature ($fdr < 0.05$).

**Figure supplement 4.** Gene expression correlations with winter temperature.

**Figure supplement 5.** Gene expression correlations with winter temperature compared to background genes.

**Source data 1.** Cold-acclimation genes and their expression cluster membership as shown in *Figure 3*.

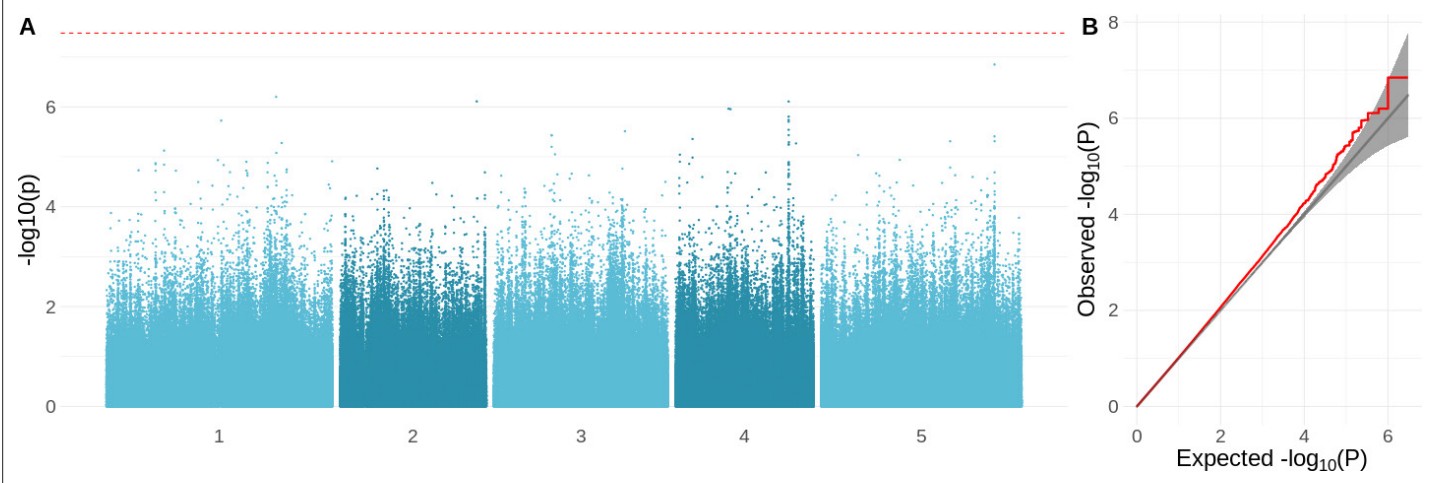

**Figure 4.** Genome-wide association study (GWAS) results for the growth rate at 16°C. (**A**) Manhattan plot showing the significance of the association between the phenotype and each of the tested SNPs ($MAF > 10\%$). The Bonferroni-corrected threshold is shown with a dashed red line. (**B**) QQ-plot showing the relation between observed and expected $-\log 10(\mathrm{p-value})$ distributions. Red line shows the observed relationship. The gray line and band show the expected relationship under the null hypothesis of no differentiation between both distributions.

The online version of this article includes the following figure supplement(s) for figure 4:

**Figure supplement 1.** Genome-wide association study (GWAS) results for the initial size, growth rate at 6°C, and the temperature response of the growth rate.

were no genome-wide significant associations (*Figure 4—figure supplement 1*). The strongest association was found for overall growth rate at 16°C (*Figure 4A*). Inflated significance levels after correcting for population structure are consistent with what we would expect from a polygenic trait (*Figure 4B*) and were also observed for the other traits, except for growth rate at 6°C (*Figure 4— figure supplement 1*). Plausible candidates within 10 kb of the most significant SNP (chr5: 23334281; $-\log_1 0(\mathrm{p-value}) = 6.85$) include *CIPK21* and *MYB36*. *CIPK21* encodes a CBL-interacting protein kinase that is upregulated in cold conditions and is involved in the salt and osmotic stress response (*Pandey et al., 2015*). MYB36 is a key regulator of root endodermal differentiation (*Liberman et al., 2015*). Slightly more distant, 22 kb away, is *COL5*, encoding a transcription factor that is part of the gene network that is regulated by AN3, a regulator of cell proliferation in leaf growth (*Vercruyssen et al., 2014*).

To test for potential polygenic adaptation, we compared the phenotypic divergence to the expected neutral genome-wide genetic divergence. This can be done using a $Q_{ST} - F_{ST}$ test (*Prout and Barker, 1993*; *Whitlock, 2008*; *Spitze, 1993*); however, this test is not well suited for species with complex population structure, and so we used a variation that was designed to detect adaptive differentiation for traits measured in structured GWAS panels (*Josephs et al., 2019*). Instead of looking at divergences between predefined populations, this method uses principal components (PCs) of the genetic relatedness matrix as axes of potential adaptive differentiation. Adaptive differentiation is then detected as a correlation between the focal phenotype and any of these relatedness PCs that is significantly different than expected under neutrality.

Adaptive differentiation was detected for initial size and for growth rate at 16°C and its temperature response. These traits show adaptive differentiation along different genetic axes (*Figure 5*). Initial size shows significant adaptive differentiation along PC6 (p − value<0.05), whereas growth rate at 16°C and its temperature response showed significant adaptive differentiation along PC5 (p − values<0.05). Adaptive differentiation was not significant along the other axes of genetic differentiation (PC1–4, PC7–10). The adaptive differentiation for initial size along PC6 seems to stem from higher initial sizes in Swedish accessions compared to central European accessions. The adaptive differentiation along PC5 seems to be driven by the lower growth rate temperature responses in Asian and northern Swedish accessions in contrast to higher growth rates in a subset of southern Swedish accessions. The accessions in our set that come from northern Sweden and Asia hail from the coldest climates. Thus, these results suggest adaptive differentiation driven by adaptation to cold winters. Given the

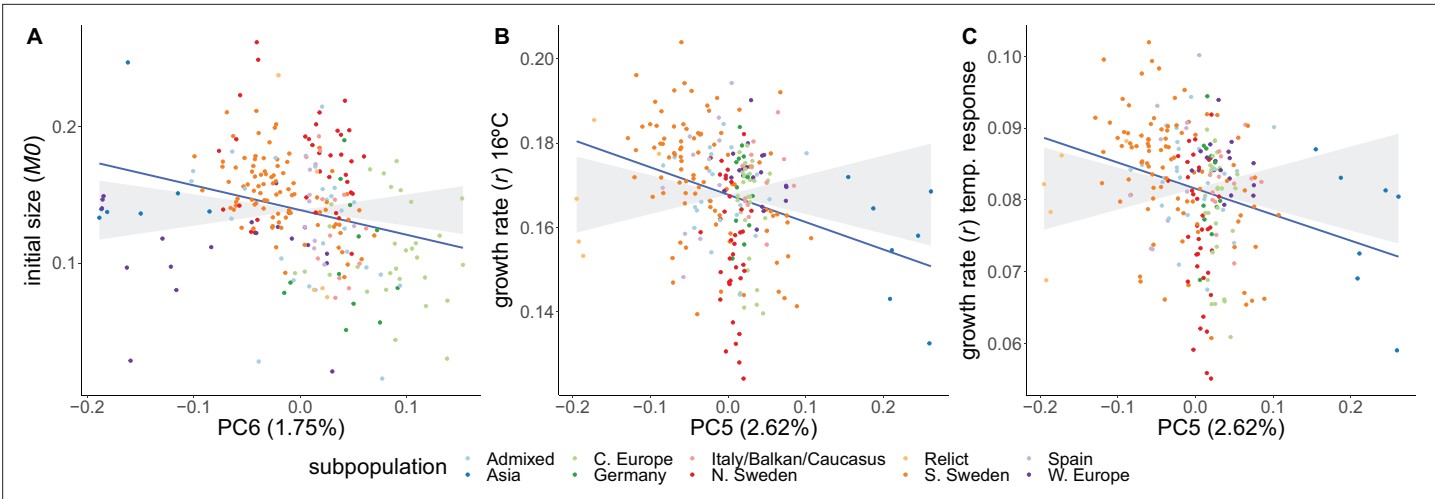

**Figure 5.** Adaptive differentiation of initial size, growth rate at 16°C, and the temperature response of growth rate along different axes of genetic differentiation. Plots represent the phenotypes and axes of genetic differentiation for which we detected significant adaptive differentiation; initial size and PC6 (**A**), growth rate in 16°C and PC5 (**B**), and the growthrate's temperature response and PC5 (**C**). Accessions are colored according to their respective admixture groups, as specified in *1001 Genomes Consortium, 2016*. The gray ribbon represents the expected correlation between phenotype and axis of genetic differentiation under neutrality with a 90% confidence interval. The neutral expectation is based on axes of genetic differentiation within populations (see 'Materials and methods' and *Josephs et al., 2019* for further details). The blue line represents the observed correlation between phenotype and axis of genetic differentiation. Percentages refer to the genetic variation explained by the respective principal component.

The online version of this article includes the following figure supplement(s) for figure 5:

**Figure supplement 1.** Adaptive differentiation of initial size, growth rate at 16°C, and the temperature response of growth rate along different axes of genetic differentiation.

seemingly strong influence from the Asian accessions, we repeated the analysis without them. Also in this analysis we detected significant adaptive differentiation ($p-values < 0.05$) for initial size, growth rate at 16°C, and its temperature response (*Figure 5—figure supplement 1*).

## Discussion

This study explores natural variation of rosette growth in nonfreezing temperatures. We detect genetic variation for the different growth parameters, and environmental correlations that suggest local adaptation. GWAS analyses reveal, not surprisingly, a polygenic trait architecture. We speculated that the slower growth measured in accessions from colder climates reflects relocation of resources from growth towards cold acclimation. Both metabolome and gene expression data are consistent with accessions from colder climates preparing for a harsh winter. In our temperature experiment, we see that the growth of northern lines is affected less than southern lines by switching from 16°C to 6°C.

Our conclusion that slower growth is likely adaptive in populations facing fiercer winters is in line with recent results of *Wieters et al., 2021*, who concluded that the reduced growth in northern lines was adaptive and not a consequence of an accumulation of deleterious mutations at the species border. If slower growth were indeed a consequence of accumulated deleterious mutations, we would expect to see slower growth also during the initial seedling establishment, which we measured here as the initial size. On the contrary, we saw a fast seedling establishment for accessions from colder regions. We speculate that the fast seedling establishment is a potential adaptation for short growth seasons, which often coincide with colder climates (high latitude or high altitude). This fast seedling establishment seems to be partly supported by larger seeds. These larger seeds may provide more nutrients to initiate faster seedling establishment, while this is of less importance in warmer climates with longer growth seasons. Further work is needed to disentangle initial growth from seed size effects and confirm that there is a causal relationship between seed size and fast seedling establishment, whether this is due to seed nutrient storage, and whether it is adaptive.

The adaptation of growth to local climates is likely to be influenced by a trade-off with cold acclimation. General growth-survival trade-offs have long been observed and are described in general ecological strategy schemes such as Grime's C-S-R triangle (*Grime, 1979*) and the leaf–height–seed scheme (*Westoby, 1998*). Specific trade-offs between growth and cold/frost survival were observed for wheat (*Hayes and Aamodt, 1927*; *Quisenberry, 1931*), alfalfa (*Castonguay et al., 2006*), *Dactylis glomerata* (*Bristiel et al., 2018*), and multiple tree species (*Koehler et al., 2012*; *Loehle, 1998*; *Molina-Montenegro et al., 2012*; *Savage and Cavender-Bares, 2013*). Here, we observed higher expression of genes involved in cold acclimation in accessions from colder regions. This is clearest at 6°C, but is also happening at 16°C, suggesting that activation of cold acclimation is stronger in accessions from cold climates and may be triggered more easily. More generally, it establishes that geographic differentiation exists in the expression of known cold-response genes. Accessions from warm climates may instead activate cold acclimation as a stress response rather than as a preventive measure. Even though this is based on a limited set of 8 accessions, metabolome measurements in all 249 accessions lead to the same conclusion. Metabolites involved in cold acclimation such as raffinose, sucrose, and proline were found in higher concentrations in accessions from colder climates (*Weiszmann et al., 2020*). We believe that accessions from colder environments are relocating more energy and resources from growth towards preparations for upcoming freezing temperatures, which is a clear example of active plasticity (*Forsman, 2015*; *van Kleunen and Fischer, 2005*). Even though we have no direct survival measurements, we speculate that this results in stronger cold acclimation and consequently increased freezing tolerance in the accessions from colder regions. Indeed, accessions originating from colder environments show increased freezing tolerance upon cold acclimation (*Zhen et al., 2011*; *Zuther et al., 2012*; *Hannah et al., 2006*; *Horton et al., 2016*). This fits with observations of northern and colder regions favoring slower growing, more stress-tolerant plants (*Vasseur et al., 2018*; *Estarague et al., 2022*). Also, biogeographic studies in *A. thaliana* found that winter temperatures are a major determinant of suitable habitats for this species (*Hoffmann, 2002*; *Yim et al., 2022*), and reciprocal transplant experiments detected an important role for freezing tolerance in fitness variation in northern sites (*Ågren and Schemske, 2012*). The high variability we observed in our data does, however, show that there is more at play than selection for cold resistance alone. What these factors are we can only speculate about. Phenotypes are shaped by a mixture of neutral and adaptive processes, with a plethora of trade-offs between traits. Investigating phenotypes at different organismal scales in specific and realistic environments will further elucidate how phenotypes and, ultimately, life history strategies are shaped. We speculate that the reduced growth plasticity observed for accessions from colder climates is due to the stronger growth reduction at 16°C in these accessions compared to accessions from warmer climates. They may well be anticipating winter, whereas accessions from warmer climates do not, and hence show a stronger difference between 16 and 6°C. Although both metabolite and gene expression data suggest an involvement of cold acclimation, we note that it is impossible to rule out that accessions from colder climates grow slower due to reduced resource efficiency as they become larger or, for example, increase leaf thickness (*Adams et al., 2016*). Further work is needed to understand the mechanism underlying the growth response observed here.

There is strong evidence from QTL mapping that genetic variation in the CBF2 gene is one of the drivers for adaptation to freezing stress (*Oakley et al., 2014*; *Gehan et al., 2015*). Here, we looked at growth phenotypes and did not detect associations with the CBF loci. In the transcriptome analysis, we did pick up a role for CBF and other known cold-acclimation genes. The most significant locus detected in our GWAS analysis (for growth rate at 16°C) lies in the vicinity of *COL5*, a gene that is part of a leaf growth regulatory network (*Vercruyssen et al., 2014*) and whose expression is induced by both cold treatment and *CBF1*, *CBF2,* and *CBF3* overexpression (*Park et al., 2015*). It is however unclear what its exact regulatory role in growth in cold conditions might be. In summary, we detected adaptive differentiation for growth between accessions from warm and cold climates. Our transcriptome data and previous metabolome data suggest that resources are relocated from growth to cold acclimation in accessions from colder regions. This allows these accessions to be fully prepared for the coming of winter.

## Materials and methods
### Plant growth and phenotyping

Seeds of 249 natural accessions (*Figure 1—source data 1*) of *A. thaliana* described in the 1001 genomes project (*1001 Genomes Consortium, 2016*) were sown on sieved (6 mm) substrate

(Einheitserde ED63). Pots were filled with 71.5 g ± 1.5 g of soil to ensure homogenous packing. The prepared pots were all covered with blue mats (*Junker et al., 2014*) to enable a robust performance of the high-throughput image analysis algorithm. Seeds were stratified (4 days at 4°C in darkness) after which they germinated and left to grow for 2 weeks at 21°C (relative humidity: 55%; light intensity: $160\,\mu\mathrm{mol\,m^{-2}\,s^{-1}}$; 14 hr light). The temperature treatments were started by transferring the seedlings to either 6 or 16°C. To simulate natural conditions, temperatures fluctuated diurnally between 16–21°C, 0.5–6°C, and 8–16°C for the 21°C initial growth conditions and the 6 and 16°C treatments, respectively (*Figure 1—figure supplement 1*). Light intensity was kept constant at $160\,\mu\mathrm{mol\,m^{-2}\,s^{-1}}$ throughout the experiment. Relative humidity was set at 55% but in colder temperatures it rose uncontrollably to maximum 95%. Daylength was 9 hr during the 16 and 6°C treatments. Each temperature treatment was repeated in three independent experiments. Five replicate plants were grown for every genotype per experiment. Plants were randomly distributed across the growth chamber with an independent randomization pattern for each experiment. During the temperature treatments (14–35 days after stratification), plants were photographed twice a day (1 hr after/before lights switched on/off), using an RGB camera (IDS uEye UI-548xRE-C; 5MP) mounted to a robotic arm. Rosette areas were extracted from the plant images using Lemnatec OS (LemnaTec GmbH, Aachen, Germany) software. Plant growth profiles were visually inspected, and datapoints with smaller rosette areas than earlier time points (negative growth) were discarded from further analyses. At 35 days after stratification, whole rosettes were harvested, immediately frozen in liquid nitrogen, and stored at –80°C until further analysis.

## Nonlinear modeling

Nonlinear modeling was used to describe plant growth in a minimum number of parameters. In a first step, we constructed a simple nonlinear model with plant size being explained by either the exponential (*Equation 1a*, *Equation 1b*) or the power-law function (*Equation 2a*, *Equation 2b*), with individual plant as a random effect for each of the model parameters; $M_0$, $r$, and $\beta$. With $\beta$ being only present in the power-law model. Models were constructed using the *nlsList* and *nlme* functions from the *nlme* package (3.1.152; *Pinheiro et al., 2021*) for R (4.0.3; *R Development Core Team, 2017*). Exponential and power-law *SelfStart* functions were used from *Paine et al., 2012*. Based on Akaike information criterion and likelihood ratio test generated by the *anova* function (*Table 1*), we decided to use the power-law model for further analyses.

$$\frac{dM}{dt} = rM \tag{1a}$$

$$M_t = M_0 e^{rt} \tag{1b}$$

$$\frac{dM}{dt} = rM^{\beta} \tag{2a}$$

$$M_t = (M_0^{1-\beta} + rt(1-\beta))^{1/(1-\beta)} \tag{2b}$$

In a second step, we constructed a model with fixed effects for the different power-law parameters. For initial size ($M_0$), we added accession as fixed effect. Temperature treatment only started from the initial time point onwards, and thus could not have an effect on the initial plant size. The growth rate, on the other hand, should be affected by temperature; therefore, we included accession, temperature, and their interaction as fixed effects for growth rate ($r$). No fixed effects were added for $\beta$. The idea here is that it is an adjustment factor for decreasing growth rates (when $\beta < 1$) with increasing plant sizes, which is general for plant growth, or at least for our data in this case. Individuals nested within experiment were added as random effects for each of the model parameters. The correlation

**Table 1.** ANOVA table for the comparison between the exponential and power-law model with degrees of freedom (df), Akaike information criterion (AIC), Bayesian information criterion (BIC), and log-likelihood (logLik) for each model.
The likelihood ratio statistic (L.ratio) and p-value are given for the likelihood ratio test that was used to compare these models.

|  | Model | df | AIC | BIC | logLik | Test | L.ratio | p-Value |
|---|---|---|---|---|---|---|---|---|
| Exponential | 1 | 6 | –668551.6 | –668488.1 | 334281.8 |  | - | - |
| Power-law | 2 | 10 | –870605.0 | –870499.1 | 435312.5 | 1 vs. 2 | 202.061 | <0.001 |

structure intrinsic to measuring the same individuals over time was accounted for by adding the first-order continuous autoregressive correlation structure (corCAR1). The estimated fixed effects of this model were then used to obtain initial size estimates for each accession and growth rate estimates for each accession in both temperatures. These estimates were used for all further analyses apart from broad-sense heritability calculations (see below). For each accession, we calculated the growth rate response as the slope between the growth rate at 6°C and the growth rate at 16°C. The slope was obtained from linear regression with the *lm* function in R (4.0.3; *R Development Core Team, 2017*) using temperature as an ordered categorical variable (6°C < 16°C).

## Climate correlations

The different phenotypes were correlated with each of the different (bio)climate variables downloaded from https://www.worldclim.org (*Fick and Hijmans, 2017*). Correlations were calculated as Pearson's correlations using the *cor* function in R (4.0.3; *R Development Core Team, 2017*). Population structure may confound the correlation between phenotype and climate. Therefore, we included additional phenotype–climate correlations with correction for population structure (*Figure 2—figure supplement 3*). For the population–structure-corrected correlations, we used a mixed-effects model as implemented in the *lmekin* function from the *coxme* (2.2.16; *Therneau, 2020*) package with phenotype as dependent variable, climate variable as fixed effects, and the kinship matrix as random effect. The kinship matrix was based on the SNPs from the 1001 genomes consortium (*1001 Genomes Consortium, 2016*) and was calculated using 'mixmogam' (https://github.com/bvilhjal/mixmogam; *Vilhjalmsson, 2019*) based on *Kang et al., 2010*. In this analysis, phenotype and climate variables were standardized, so that regression coefficients were comparable to correlation coefficients. Even though the strength and significance of the correlations weaken upon population structure correction, the growth parameters still demonstrate the same pattern, being most strongly correlated with winter temperatures.

## Seed size correlations

We used the seeds produced by *Kerdaffrec et al., 2016* and limited our measurements to the set of 123 Swedish accessions that overlapped with our growth dataset. After seed stratification for four days at 4°C in darkness, mother plants were grown for 8 weeks at 4°C under long-day conditions (16 hr light; 8 hr dark) to ensure proper vernalization. Temperature was raised to 21°C (light) and 16°C (dark) for flowering and seed ripening. Seeds were kept in darkness at 16°C and 30% relative humidity, from the harvest until seed size measurements. For each genotype, three replicates were pooled and about 200–300 seeds were sprinkled on 12 × 12 cm², transparent Petri dishes. Image acquisition was performed as described in *Exposito-Alonso et al., 2018* by scanning dishes on a cluster of eight Epson V600 scanners. The resulting 1200 dpi .tiff images were analyzed with the ImageJ software (2.1.0/1.53c). Images were converted to eight-bit binary images and thresholded with the *setAuto-Threshold*("*Defaultdark*") command, and seed area was measured in mm² by running the *Analyse Particles* command (inclusion parameters: $size = 0.04 − 0.25$). All scripts used for image processing are available at https://github.com/vevel/seed_size; *Kerdaffrec, 2022*. The variance decomposition for initial size into variance explained by winter temperature and seed size was done with a random-effect model where initial size was explained by winter temperature and seed size, using the *lmer* and *VarCorr* functions from the *lme4* package (1.1.27.1; *Bates et al., 2015*) in R (4.0.3; *R Development Core Team, 2017*). The seed size-corrected correlation between initial size and winter temperature was estimated with the *lme* function from the *nlme* package (3.1.152; *Pinheiro et al., 2021*) in R (4.0.3; *R Development Core Team, 2017*), and the correction for seed size was done by including seed size as a random effect.

## Transcriptome profiling

35 days after stratification, rosette tissue of all plants were harvested and flash frozen in liquid nitrogen. Random samples from each replicate experiment for both temperatures were taken for eight accessions to profile the transcriptome with RNA-sequencing. The eight accessions were selected to represent the climatic variation in the full panel (*Figure 3—figure supplement 1*, *Figure*

1—source data 1). Total RNA was extracted using the KingFisher Duo Prime System (Thermo Fisher Scientific) together with a high-performance RNA bead isolation kit (Molecular Biology Service, VBC Core Facilities, Vienna). To determine the quantity of RNA, we used Fluorometer Qubit 4 (Invitrogen) and Qubit RNA BR Kit (Invitrogen). For each sample, 1 μg of total RNA was treated with the poly(A) RNA Selection Kit (Lexogen) and eluted in 12 μl of Nuclease-Free Water. Libraries were prepared according to the manufacturer's protocol in NEBNext Ultra II Directional RNA Library Prep Kit (New England Biolabs) and individually indexed with NEBNext Multiplex Oligos for Illumina (New England Biolabs). The quantity and quality of each amplified library were analyzed by using Fragment Analyzer (Agilent) and HS NGS Fragment Kit (Agilent). Libraries were sequenced with an Illumina HiSeq2500 in paired-end mode with read length of 125 bp. Sequencing was performed by the Next Generation Sequencing Facility at Vienna BioCenter Core Facilities (VBCF), member of the Vienna BioCenter (VBC), Austria. Samples were distributed over four independent libraries. This was due to failed samples that needed replacement. Detailed info on which samples belong to which library is listed in SRA (https://www.ncbi.nlm.nih.gov/sra/PRJNA807069). Gene expression was quantified by using quasi-mapping in salmon (1.2.1; *Patro et al., 2017*). The salmon indices were built separately for each accession as we incorporated the SNP variation from the *1001 Genomes Consortium, 2016* into the reference transcriptome. The heatmap was built using *pheatmap* (1.0.12; *Kolde, 2019*) in R (4.0.3; *R Development Core Team, 2017*). The clustering was done by complete clustering on Euclidean distances for both rows and columns. The gene clusters were defined by cutting the dendrogram in seven groups. With a -test, we tested for overrepresentation of a temperature effect on the expression of the 251 selected cold-acclimation genes ($df = 1$) compared to the remaining 18,784 background genes. We used the *chisq.test* function in R (4.0.3; *R Development Core Team, 2017*). Differential expression analysis was conducted with the *DESeq2* package (1.30.0; *Love et al., 2014*) in R (4.0.3; *R Development Core Team, 2017*). A full model was used, with expression depending on *replicate + accession + temperature + replicate : temperature + accession : temperature* , after which significance of each model coefficient was defined with a negative binomial Wald test. Differential expression for each accession was then extracted by specifying the respective contrasts using the *lfcShrink* function in *DESeq2* with the adaptive shrinkage estimator (*Stephens, 2017*). Genes were considered differentially expressed when the adjusted p-value was <0.05.

## Metabolome profiling

Besides transcriptome profiling of eight accessions, we also conducted metabolome profiling on all 249 accessions. Samples for metabolome measurements were taken from the same experiments described in this study and, just like the transcriptome samples, were taken 35 days after stratification. Results and detailed methodology are described in *Weiszmann et al., 2020*.

## Broad-sense heritabilities

Broad-sense heritabilities ($H^2$) were calculated as the ratio between phenotypic variation explained by genotype ($V_g$) and the total phenotypic variation ($V_p$), which is the sum of $V_g$ and phenotypic variation explained by environment ($V_e$). These variances were obtained from a mixed model by including accession as a random effect (estimate for $V_g$). Because our accession estimates were corrected for experiment effects, we removed the variance explained by experiment by including experiment as a fixed effect. In order to estimate the variance within each accession ($V_e$), the dependent variables in this model were the growth parameter estimates for each individual plant, in contrast to the estimates for each accession that were used in all other analyses. These individual plant estimates were obtained from the same model, but took the random effects into account. The variance explained by accession was then taken as an estimate for $V_g$, the residual variance was taken as $V_e$. For initial size, we calculated heritability over all experiments, and growth rate heritabilities were calculated for each temperature independently. The mixed model was constructed with the *lmer* function in the *lme4* package (1.1.27.1; *Bates et al., 2015*) in R (4.0.3; *R Development Core Team, 2017*).

## Genome-wide association mapping

Genome-wide association mapping was done for each of the growth parameters in both temperatures and also the temperature response for the growth rate. We used a mixed model with phenotype as dependent variable, genotype as fixed effect, and genetic relatedness as random factor. Nonimputed

SNPs obtained from the *1001 Genomes Consortium, 2016* were used as genotypes. This model was run in GEMMA (0.98.3; *Zhou and Stephens, 2012*), with kinship matrix calculated as the centered relatedness matrix, as implemented in GEMMA.

### Testing for adaptive differentiation

Adaptive differentiation was tested with the method described by *Josephs et al., 2019* and the accompanying quaint package (0.0.0.9; https://github.com/emjosephs/quaint; *Josephs, 2020*) in R (4.0.3; *R Development Core Team, 2017*). The kinship matrix was calculated using the *make_k* function in the quaint package. Genetic PCs were then calculated from the eigen decomposition of the kinship matrix. Adaptive differentiation of each phenotype along the first 10 PCs was tested with the *calcQpc* function in the quaint package. PCs 11–248 were used to build the expected phenotypic differentiation under neutrality.

### Supplementary information

Scripts can be found at https://github.com/picla/growth_16C_6C/; *Clauw, 2022*. Scripts for seedsize analysis can be found at https://github.com/vevel/seed_size; *Kerdaffrec, 2022*.

All RNA-sequencing were uploaded to SRA under http://www.ncbi.nlm.nih.gov/bioproject/807069. All generated phenotyping data are filed under 10.5281/zenodo.6076948.

### Acknowledgements

We thank past and current members of the Nordborg group for their help in setting up and harvesting these experiments. Thanks also to Daniele Filiault for her valuable comments on the manuscript.

## Additional information

### Competing interests

Magnus Nordborg: Reviewing editor, *eLife*. The other authors declare that no competing interests exist.

### Funding

No external funding was received for this work.

### Author contributions

Pieter Clauw, Conceptualization, Data curation, Formal analysis, Investigation, Writing - original draft; Envel Kerdaffrec, Resources; Joanna Gunis, Ilka Reichardt-Gomez, Viktoria Nizhynska, Stefanie Koemeda, Investigation; Jakub Jez, Supervision, Investigation; Magnus Nordborg, Conceptualization, Supervision, Funding acquisition, Project administration, Writing - review and editing

### Author ORCIDs

Pieter Clauw ⬤ http://orcid.org/0000-0002-9677-8727
Envel Kerdaffrec ⬤ http://orcid.org/0000-0001-8667-6850
Ilka Reichardt-Gomez ⬤ http://orcid.org/0000-0001-7964-9352
Jakub Jez ⬤ http://orcid.org/0000-0002-6481-4383
Magnus Nordborg ⬤ http://orcid.org/0000-0001-7178-9748

### Decision letter and Author response

Decision letter https://doi.org/10.7554/eLife.77913.sa1
Author response https://doi.org/10.7554/eLife.77913.sa2

## Additional files

### Supplementary files
• MDAR checklist

## Data availability

Scripts can be found at https://github.com/picla/growth_16C_6C/, copy archived at swh:1:rev:f-32793d07a068a6c49c876536f684ffebdcd9e6b. Scripts for seedsize analysis can be found in https://github.com/vevel/seed_size, copy archived at swh:1:rev:748523031aa64601720328ab49e02009a6fb70da. All RNA-sequencing were uploaded to SRA under http://www.ncbi.nlm.nih.gov/bioproject/807069. All generated phenotyping data are filed under https://doi.org/10.5281/zenodo.6076948.

The following dataset was generated:

| Author(s) | Year | Dataset title | Dataset URL | Database and Identifier |
|---|---|---|---|---|
| Clauw P | 2022 | Arabidopsis transcriptome in 16C and 6C | https://www.ncbi.nlm.nih.gov/bioproject/PRJNA807069 | NCBI BioProject, PRJNA807069 |

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
