## [Editor Report]

The article combines genetic and phenotypic approaches to show convincing evidence of local adaptation in early vegetative growth of *Arabidopsis* lineages sampled from a wide range of locations. The authors show larger initial size and slower growth of northern accessions compared to southern accessions when exposed to cold temperatures, suggesting that northern accessions potentially reallocate resources for winter survival. This study is commendable for its scope and comprehensive analysis of local adaptation of a highly polygenic trait in a model weed.

---

## [Decision Letter]

**Decision letter after peer review:**

Thank you for submitting your article "Locally adaptive temperature response of vegetative growth in *Arabidopsis thaliana*" for consideration by *eLife*. Your article has been reviewed by 3 peer reviewers, including Regina S Baucom as Reviewing Editor and Reviewer #1, and the evaluation has been overseen by Christian Landry as the Senior Editor.

Essential revisions:

(1) The work needs significant re-framing for broad relevance in the introduction. As is currently written, the manuscript is perhaps of more interest to a plant biology journal. Re-framing will be essential for *eLife*'s broad audience. Each reviewer provides suggestions to this end.

(2) Each major conclusion needs better justification (ie need to see the results of analysis, F-values, P-values). Many of the conclusions were presented without such results; a supplemental document is likely needed in this regard. Reviewers felt results from the phenotype data, metabolome data, transcriptome data etc all were a bit weak in this regard.

(3) A number of conclusions drawn were questioned by reviewers, with suggestions for re-analysis and clarity surrounding analyses. Each of these in turn should be considered.

*Reviewer #1 (Recommendations for the authors):*

I enjoyed reading this work; it is a very extensive set of data nicely pulled together into a story and it will add a useful examination of plant growth adaptation to the literature. I list areas of improvement below according to line number.

Introduction

1. Lines 29-46 lack a clear and novel hypothesis that the authors are addressing; I suggest reframing here in terms of life-history adaptation and trade-offs between growth for reproduction and survival.

2. Line 48 for clarity, remove ‘upcoming’ and ‘consequently’.

Results

3. Presentation of sample sizes is sometimes unclear. Line 71 rephrase to 'Our replicated experiment yielded dense…' and '(5 replicates X 249 accessions X 2 treatments X 3 experiments)'.

4. In Line 74-80, unclear why authors chose power law here – is this based on their own data or previous analyses? Methods make it apparent that authors have modeled this themselves, yet there is no presentation of model fit analysis in the results. This section really needs data in support of conclusions – report AIC, etc, explaining why power law is appropriate. Or, develop this in supplemental text.

5. Lines 91-94. The authors jump back and forth between using accession as a fixed effect in some analyses, and then random effect in others. Explanation of these choices would improve the manuscript.

6. Lines 105-115. This is a pretty significant section yet the data are not shown – estimates of the strength of the correlation between growth parameters and winter temperature if authors assert 'strongly correlated' should be shown in text as a range. Figures S1 and S2 are very nicely displayed but the assertion of a strong correlation should have a number associated with it so that readers can form their own conclusions. Typically 'strong' correlations are 0.5+ but often 0.7+. Lower than 0.4 are most often considered moderate.

7. Lines 117-145. It appears from the regressions that Asian populations are doing a lot of the work. This is certainly biologically relevant (although not discussed enough in the discussion!) so I am not suggesting the authors should remove these populations but they should report how much of their overall conclusions are driven by the populations experiencing the coldest temperatures. How much are these populations driving overall trends?

Additionally, the data in this section are discussed predominantly as the outcome of regressions, with R2 values reported on the figure. In the methods, it is clear that authors modeled effects using fixed and mixed analyses of variance, yet none of these results are apparent in text or in supplemental. Readers, myself included, want to see effect sizes, F- and p-values associated with the claims made; without them, the work appears analyses-lite.

How much of the variation in size was due to seed size? Numbers are necessary to back up claims such as 'seed size alone' etc. What proportion of the variance is leftover once seed size is corrected? I would suggest modeling seed size as a fixed covariate, unclear why the choice to model it as a random effect was made.

Line 138. Why aren't resources limiting? Do the authors mean resources in this particular growth chamber study aren't limiting?

Lines 141-145. The conclusion here is hard to accept given that the Swedish populations show a huge range in response to temperature changes.

8. Lines 147+. A metabolome experiment is mentioned yet there was not a metabolomics section in the methods, nor was there clear data analysis of a metabolome experiment. I assume authors are referencing a previously published dataset but details should be present in this work as well.

9. Line 196. Remove 'so-called'.

10. Lines 208+. The influence of the Asian populations should be assessed in the test of adaptive differentiation.

11. Figure 2. This figure is difficult to interpret. I like the idea of having a conceptual map of the experimental design, but the 'seedling establishment' banner is odd (perhaps put this text on the x-axis during seedling establishment phase and not within the temps on the y axis?). The inlay showing the light/temp acclimation period is hard to interpret, why are the y-axes off-kilter from the overall figure? Y-axis should be labeled 'Temperature'.

Discussion

12. Line 227 'Despite high plasticity' <- unclear where this conclusion comes from – current data of temperature response? If so, then plasticity needs to be discussed at more length in the results. Or, if authors are concluding this from previously published work, a citation is needed.

13. Lines 237-239. This statement is difficult to accept given the wide variation among Swedish populations.

14. Lines 245-255. The discussion of seedling establishment is a bit long especially since I didn't see seedling establishment data presented per se. How was fast seedling establishment measured and was it compared between northern and southern latitudes?

Methods.

15. Most comments above on data presentation can be applied to the methods.

Line 419 'each replicate' is unclear – I believe the authors meant replicate experiment here not replicate plants. The overall transcriptome sampling is unclear – how many total libraries were developed?

Was there a metabolome experiment? If so it should be explained briefly in the methods.

*Reviewer #2 (Recommendations for the authors):*

1. The metabolome analysis is mentioned in the discussion (lines 270-272) but not reported in the results.

2. There are a number of crucial pieces of information missing from the presentation of the gene expression results. How were the dendrograms in figure 5 made? How were these clusters generated? Is this figure only for the 251 genes or for a larger set of genes? What was the background set of genes for the Chi-square reported in line 161? Please provide additional details to answer these questions.

3. The information presented in Figure S5 seems very relevant and interesting to the main results, but this figure could be improved. It is very hard right now to read the trend line between the temperature of origin and expression for each cluster/treatment, and this would be clearer if the lines were a contrasting color.

4. We were confused about the statistical tests reported in Figure S5, since they do not appear to be described in the methods. It seems like this figure reports the correlation between the temperature of origin and expression for every gene in the cluster. This analysis would be pseudoreplication since the expression of different genes from the same genotype is not independent. A more appropriate analysis would be to estimate the correlation of temperature of origin with the mean expression or eigengene for all genes in the cluster. Alternatively, the correlation between the temperature of origin and expression could be independently calculated for each gene in a cluster, and then the number of significant correlations and/or mean slope of these correlations within each cluster could be reported.

5. In the paragraph beginning in line 261, it is clear how a slower growth rate in accessions from cold regions at 6C is consistent with cold acclimation literature but not how the reduced growth rate plasticity in these accessions is connected to the literature. The discussion could be added to explain why the accessions from cold regions also grow more slowly when never exposed to cold (expand on line 138) and why they might have less plasticity in the growth rate.

Additional line comments:

6. The paragraph starting on line 47 could use more context for how the information relates to the broader argument.

7. Line 123 should cite where the seed size information is from.

8. It would be useful to report the p-value and effect size of the "most significant" association (line 179).

9. What are the "inflated significance levels" referred to on lines 180-181?

10. Should include statistics to justify the model choice in lines 348-351.

11. Lines with potential typos: 344, 360, 462.

12. Figures should have larger axes and labels, and all panels should be labeled within a figure. For example, label the panels in Figures3 and 4.

13. In figure 4, what impact does the x-axis outlier (coldest temp) have on the patterns we see? It would be nice to know that the outlier is not driving the trends and to have an explanation of why that accession is much colder than the rest (for example, where is this accession from?).

14. The Qst vs Fst analysis was used appropriately but there could be an additional explanation for how to interpret figure 7 in regards to how the gray fields are calculated

15. Unclear what "high plasticity" means in line 227.

*Reviewer #3 (Recommendations for the authors):*

Line 35: remove so-*

Line 47-58: the emphasis on CBFs is maybe excessive here as you don't provide compelling evidence for their involvement. Maybe swap for an introduction of the main ecological strategy of Nordic vs Southern adapted accessions?

Line 74-82: Provide the AIC/BIC/deviance or RMSE to justify the use of power vs exponential growth model in materials, but keep it simple here so that the interpretation of the parameter is clearer.

Figure 2: Upon vernalisattion? I believe you mean cold stratification?

Line 94: This is peripheral but I wonder if there is no biological insight to be gained from fitting a β for each accession. Wouldn't this be informative about the rate of resource accumulation?

Line 103-128: I understand that some metrics are provided in the figures associated with these sections but it would be clearer to provide some quantitative support (and tests) in the main text.

Line 147: how does the metabolite measurement fit?

Line 154: which accessions? Give the rationale for selection based on Figure S4.

Line 162: the null hypothesis of the test is not known.

Line 171-174: I find this argument circular: these genes were identified as differentially expressed in the very same accession I presume (although this is not explicitly stated in the paper).

Line 134: relocation of resources towards… I think there could be some information in analysing β's inferred for each accession here.

Line 360: typo

Line 372: Put the rest of the section in the heritability section.

Line 392: grammar issue.

Line 462: grammar again.

[Editors’ note: further revisions were suggested prior to acceptance, as described below.]

Thank you for resubmitting your work entitled "Locally adaptive temperature response of vegetative growth in *Arabidopsis thaliana*" for further consideration by *eLife*. Your revised article has been evaluated by Christian Landry (Senior Editor) and a Reviewing Editor.

The manuscript has been improved but there are some remaining issues that need to be addressed, as outlined below:

*Reviewer #2 (Recommendations for the authors):*

The clarity of the manuscript has been improved particularly in the descriptions of the methods and figure design. A few additional analyses to explore subsets of the data that drive trends in growth rate are included which strengthens the overall conclusion that temperature response of vegetative growth is locally adaptive. However, the manuscript has not addressed two of our initial concerns about the broad framing of the manuscript and the gene expression analyses.

1. The manuscript has been reframed in the context of ecological and life history strategies with a focus on the growth and survival tradeoff, based on feedback from us and other reviewers. While the growth rate results are potentially consistent with what is expected for a growth/survival tradeoff, there is no evidence for the other side of the tradeoff, survival. The manuscript should make this clear. In addition, the current framing includes references to active and passive plasticity, but we are not convinced that faster growth of accessions from warm temperatures "proves" that slow growth of accessions from cold temperatures is active plasticity (lines 125-127) as opposed to another explanation such as less efficient use of resources in accessions from cold temperatures at a later life stage. Additionally, the manuscript notes that "plant growth is known to slow down with increasing size" when justifying its model choice for estimating growth rates. With this in mind, is it a surprise that the populations with initial larger rosettes now show slower growth?

2. The gene expression results should be better integrated to support the conclusion that growth rate response is locally adapted. The main results described in the text are (1) the cold acclimation genes are more likely to be differentially expressed between temperatures compared to background genes. This is unsurprising since these genes were initially identified as being differentially expressed in the cold. (2) Gene expression clustering separates accessions from warmer and colder climates and the expression of some clusters is correlated with the temperature of origin. It is unclear if this pattern differs from background genes or the trait divergence expected under neutrality. (3) Genes that were up-regulated in cold (from published experiments) are also generally upregulated in accessions from cold climates. This last result is potentially the clearest link to the adaptive story, but it is not emphasized here and it's also unclear to us if upregulating these genes is a sign of adaptive or maladaptive plasticity. Overall, there is not a clear link between these results and the conclusion about adaptation to cold temperatures.

In addition, while Figure 3 —figure supplement 3 does address our concern about pseudoreplication, Figure 3 —figure supplement 2 still reports misleading p values that should not be included due to pseudoreplication.

Line Comments:

The statements made in lines 41-43 should have citations.

Lines 154-156 seem circular.

Line 181 change "both" to "all".

Line 198-199. The phrase "across the genome we detect adaptive differentiation for certain growth parameters." seems unsupported by the data. While the Qst-Fst results show evidence that phenotypic variation is adaptive, there is still no evidence that this trait is shaped by loci 'across the genome'.

Line 273 should read 160 µmolm−2s-1?

It is not clear how the growth rate temperature response was calculated. Line 91-92 suggests that it is "the slope between the growth rate at 16ºC and 6ºC" but examining specific points in figure 2B, C, and D makes it look like something else is going on. For example, the far left point has a growth rate of ~0.16 in 16C and ~0.052 in 6C, but the response is ~0.072 when we would expect something closer to 0.1 based on the text.

Please include percent variance explained on figures with PCs as axes (Figure 5 and Figure 5 supplement).

Figure 1—figure supplement 1 is somewhat confusing with the y axis referring to the experiment timeline on the x-axis and to the inset graph of a single day.

The inclusion of the previously published metabolite data is still somewhat confusing. The results report that metabolite measurements were different between accessions from different climates (line 137) but these results are not presented in a figure anywhere. If the authors want to present previously published data that has been reanalyzed to this paper in a figure, that would be fine. However, without that, it seems like this statement should be moved to the discussion.

---

## [Author Response]

Essential revisions:1) The work needs significant re-framing for broad relevance in the introduction. As is currently written, the manuscript is perhaps of more interest to a plant biology journal. Re-framing will be essential for eLife's broad audience. Each reviewer provides suggestions to this end.

We agree with this point, and have revised both Introduction and Discussion accordingly. See below for details.

2) Each major conclusion needs better justification (ie need to see the results of analysis, F-values, P-values). Many of the conclusions were presented without such results; a supplemental document is likely needed in this regard. Reviewers felt results from the phenotype data, metabolome data, transcriptome data etc all were a bit weak in this regard.

This was mostly due to poor presentation, and we have revised accordingly throughout. In particular, we have expanded Materials and methods.

3) A number of conclusions drawn were questioned by reviewers, with suggestions for re-analysis and clarity surrounding analyses. Each of these in turn should be considered.

Again we agree, and we have provided additional analyses to address these concerns. Most importantly, we reanalyzed the data with outlier groups removed to confirm that the conclusions still hold. Please see below for details.

Reviewer #1 (Recommendations for the authors):I enjoyed reading this work; it is a very extensive set of data nicely pulled together into a story and it will add a useful examination of plant growth adaptation to the literature. I list areas of improvement below according to line number.Introduction1. Lines 29-46 lack a clear and novel hypothesis that the authors are addressing; I suggest reframing here in terms of life-history adaptation and trade-offs between growth for reproduction and survival.

Thank you for this suggestion. We have incorporated stronger links to the life-history/ecological strategy literature, in both the Introduction and the Discussion.

2. Line 48 for clarity, remove ‘upcoming’ and ‘consequently’.

The sentence has been rewritten.

Results3. Presentation of sample sizes is sometimes unclear. Line 71 rephrase to 'Our replicated experiment yielded dense…' and '(5 replicates X 249 accessions X 2 treatments X 3 experiments)'.

Done

4. In Line 74-80, unclear why authors chose power law here – is this based on their own data or previous analyses? Methods make it apparent that authors have modeled this themselves, yet there is no presentation of model fit analysis in the results. This section really needs data in support of conclusions – report AIC, etc, explaining why power law is appropriate. Or, develop this in supplemental text.

This section has been expanded, and the revised manuscript contains the Anova table (Table S1) with the results of the comparison between an exponential and a power-law model. This table gives degrees of freedom, AIC, BIC and the log-likelihood for each model, and the likelihood ratio and p-value for the comparison of both models. We mention in the text that the power-law model is a standard model for these kinds of data.

5. Lines 91-94. The authors jump back and forth between using accession as a fixed effect in some analyses, and then random effect in others. Explanation of these choices would improve the manuscript.

This is a misunderstanding, again presumably due to unclear writing. Accession was consistently used as a fixed effect in the main power-law model. This model produced estimates for the different growth parameters for each accession.

We suspect the confusion comes from the description of the broad sense-heritabilities. To calculate these, we used growth parameter estimates for each individual plant. These estimates were obtained from the same power-law model, but estimates now take the random effect for individual plants into account, so that we could partition the variance into genetic and “environmental” variance and calculate heritability.

We have revised the two relevant Materials and methods sections (“Non-linear modeling” and “Broad-sense heritabilities”) to make this clearer.

6. Lines 105-115. This is a pretty significant section yet the data are not shown – estimates of the strength of the correlation between growth parameters and winter temperature if authors assert 'strongly correlated' should be shown in text as a range. Figures S1 and S2 are very nicely displayed but the assertion of a strong correlation should have a number associated with it so that readers can form their own conclusions. Typically 'strong' correlations are 0.5+ but often 0.7+. Lower than 0.4 are most often considered moderate.

The correlation coefficients for growth parameters and winter temperature are actually given in Figure 4. We have added a figure reference to the text, and also toned down the language. Our main point is that growth parameters are most strongly correlated with winter temperature.

7. Lines 117-145. It appears from the regressions that Asian populations are doing a lot of the work. This is certainly biologically relevant (although not discussed enough in the discussion!) so I am not suggesting the authors should remove these populations but they should report how much of their overall conclusions are driven by the populations experiencing the coldest temperatures. How much are these populations driving overall trends?

Excellent question. As noted above, we tried this, and it turns out not to matter. We did a similar analysis for the Swedish accessions with similar results – see comments below. We note this in the text, and provide supplemental figures (Figure 2—figure supplement 4,6).

Additionally, the data in this section are discussed predominantly as the outcome of regressions, with R2 values reported on the figure. In the methods, it is clear that authors modeled effects using fixed and mixed analyses of variance, yet none of these results are apparent in text or in supplemental. Readers, myself included, want to see effect sizes, F- and p-values associated with the claims made; without them, the work appears analyses-lite.

The mixed-model analysis was only used for the population structure-corrected climate correlations in Figure 2—figure supplement 3. The other analyses are standard regressions. We adjusted the Materials and methods section on climate correlations to clarify this.

How much of the variation in size was due to seed size? Numbers are necessary to back up claims such as 'seed size alone' etc. What proportion of the variance is leftover once seed size is corrected? I would suggest modeling seed size as a fixed covariate, unclear why the choice to model it as a random effect was made.

We added a variance decomposition to see how much of the variation in initial size is explained by winter temperature and by seed size (32.7% and 11.9% of the variation, respectively). Together with the significant correlation when correcting for seed size, this shows that winter temperature is still strongly correlated with initial size when seed size is taken into account.

We chose to add seed size as a random effect since we only wanted an estimate for the correlation between initial size and winter temperature.

Line 138. Why aren't resources limiting? Do the authors mean resources in this particular growth chamber study aren't limiting?

Yes, this is a reference to the higher growth rates of the accessions from warmer climates. Those higher growth rates prove that resources are not limited and thus accessions with slower growth could in principle also exhibit similarly high growth. We clarified this in the text.

Lines 141-145. The conclusion here is hard to accept given that the Swedish populations show a huge range in response to temperature changes.

The Swedish accessions are indeed showing a very large spread for the temperature response of the growth rate. We added a supplementary figure (Figure 2—figure supplement 6) showing the coefficient of variance for each of the subpopulations and its correlation with winter temperature, plots of correlation between the growth rate's temperature response and winter temperature for all except the Swedish subpopulations, and the same correlation for only the Swedish subpopulations. These plots show that accessions from colder temperatures still have lower plasticity for this trait, also within the Swedish subpopulations the conclusion still holds.

8. Lines 147+. A metabolome experiment is mentioned yet there was not a metabolomics section in the methods, nor was there clear data analysis of a metabolome experiment. I assume authors are referencing a previously published dataset but details should be present in this work as well.

The metabolome experiment that we mention was indeed described in a previous study. We added a small selection describing this in Materials and methods, with a clear reference to the respective paper.

9. Line 196. Remove 'so-called'.

Done

10. Lines 208+. The influence of the Asian populations should be assessed in the test of adaptive differentiation.

Yes, thanks for this suggestion. We assessed the influence of the Asian accessions on the detected adaptive differentiation by excluding them. It is important to note here that the principal components needed to be recalculated after doing this, and are thus not the same as the principal components in the original analysis.

In summary, we still detect adaptive differentiation (Figure 5—figure supplement 1). For initial size there is significant adaptive differentiation along 3 different axes of genetic differentiation (PC1, PC5 and PC9). PC1 is mainly splitting northern Swedish and to a lesser extent also southern Swedish accessions from the rest, with both showing higher initial sizes compared to the other accessions. PC5 is differentiating southern Swedish and to some extent northern Swedish from central European accessions. PC9 is more enigmatic, splitting up two subgroups of the western European accessions. For both growth rate at 16ºC and its temperature response we detect adaptive differentiation along PC5, splitting up southern Swedish, northern Swedish and central European accessions. Linking back to the initial analysis, we can say that there is adaptive differentiation for initial size, growth rate at 16ºC and its temperature response. Asian accessions are driving part of this adaptive differentiation, however, also for the Swedish accessions we detect significant signals of being differently adapted in terms of these growth parameters.

This analysis is added to the manuscript (Figure 5—figure supplement 1, line 267-271).

11. Figure 2. This figure is difficult to interpret. I like the idea of having a conceptual map of the experimental design, but the 'seedling establishment' banner is odd (perhaps put this text on the x-axis during seedling establishment phase and not within the temps on the y axis?). The inlay showing the light/temp acclimation period is hard to interpret, why are the y-axes off-kilter from the overall figure? Y-axis should be labeled 'Temperature'.

We have implemented your suggestions for improving the figure, however, since reviewer #3 did not find this figure very informative, we have moved it to supplement.

Discussion12. Line 227 'Despite high plasticity' <- unclear where this conclusion comes from – current data of temperature response? If so, then plasticity needs to be discussed at more length in the results. Or, if authors are concluding this from previously published work, a citation is needed.

Correct, we don't draw conclusions about plasticity. We removed this statement since it has no real use and will cause confusion.

13. Lines 237-239. This statement is difficult to accept given the wide variation among Swedish populations.

Since the correlations without the Swedish accessions are still holding (see earlier comment), we think that this claim can still be made.

14. Lines 245-255. The discussion of seedling establishment is a bit long especially since I didn't see seedling establishment data presented per se. How was fast seedling establishment measured and was it compared between northern and southern latitudes?

We clarified that we measured seedling establishment based on the initial size, one of the growth parameters from the power-law model.

Methods.15. Most comments above on data presentation can be applied to the methods.Line 419 'each replicate' is unclear – I believe the authors meant replicate experiment here not replicate plants. The overall transcriptome sampling is unclear – how many total libraries were developed?

Adjusted 'replicate' to 'replicate experiment'. Information about libraries has been added.

Was there a metabolome experiment? If so it should be explained briefly in the methods.

Materials and methods now contains a section on the metabolome measurements.

Reviewer #2 (Recommendations for the authors):1. The metabolome analysis is mentioned in the discussion (lines 270-272) but not reported in the results.

The metabolome analysis had been published. For completeness, we added a specific section in Materials and methods. See also earlier comments by Reviewer #1.

2. There are a number of crucial pieces of information missing from the presentation of the gene expression results. How were the dendrograms in figure 5 made? How were these clusters generated? Is this figure only for the 251 genes or for a larger set of genes? What was the background set of genes for the Chi-square reported in line 161? Please provide additional details to answer these questions.

Yes: we have clarified the gene expression results. The caption of Figure 5 (now Figure 3) now explicitly states that it contains the 251 cold acclimation genes. Further details on the clustering method, chi-square test and differential expression analysis are added to Materials and methods.

3. The information presented in Figure S5 seems very relevant and interesting to the main results, but this figure could be improved. It is very hard right now to read the trend line between the temperature of origin and expression for each cluster/treatment, and this would be clearer if the lines were a contrasting color.

Due to image conversion, the contrast in the figure (now Figure 3—figure supplement 2) in the submitted pdf is pretty bad indeed, and we apologize for not noticing this. The original figure does provide a good contrast between the individual gene lines and the trend line. We further improved the contrast by adding a black shadow around the trend lines. This has been fixed.

4. We were confused about the statistical tests reported in Figure S5, since they do not appear to be described in the methods. It seems like this figure reports the correlation between the temperature of origin and expression for every gene in the cluster. This analysis would be pseudoreplication since the expression of different genes from the same genotype is not independent. A more appropriate analysis would be to estimate the correlation of temperature of origin with the mean expression or eigengene for all genes in the cluster. Alternatively, the correlation between the temperature of origin and expression could be independently calculated for each gene in a cluster, and then the number of significant correlations and/or mean slope of these correlations within each cluster could be reported.

Agreed. We have added a supplemental figure (Figure 3—figure supplement 3) that shows the proportion of genes in each cluster that significantly correlates with winter temperature.

5. In the paragraph beginning in line 261, it is clear how a slower growth rate in accessions from cold regions at 6C is consistent with cold acclimation literature but not how the reduced growth rate plasticity in these accessions is connected to the literature. The discussion could be added to explain why the accessions from cold regions also grow more slowly when never exposed to cold (expand on line 138) and why they might have less plasticity in the growth rate.

We have added a small paragraph to the Discussion where we speculate about the meaning of the reduced growth plasticity in accessions from cold regions.

Additional line comments:6. The paragraph starting on line 47 could use more context for how the information relates to the broader argument.

We added an explanation that it is not clear how natural variation in cold acclimation is in a trade-off with growth responses in cold temperatures. We do think it is important to have a brief introduction to cold acclimation and its regulators, since that information was used to select the candidate genes involved in cold acclimation.

7. Line 123 should cite where the seed size information is from.

Seed size measurements came from other experiments done in the group and were never published. Methodology is described in Materials and methods and we made clear in the text that the data comes from unpublished experiments.

8. It would be useful to report the p-value and effect size of the "most significant" association (line 179).

P-value and effect size are added to the results.

9. What are the "inflated significance levels" referred to on lines 180-181?

The text now refers explicitly to the QQ-plot in Figure 4B.

10. Should include statistics to justify the model choice in lines 348-351.

We added the ANOVA table on which we based our model choice. See also comment by reviewer #1.

11. Lines with potential typos: 344, 360, 462.

Fixed: thanks for pointing these out.

12. Figures should have larger axes and labels, and all panels should be labeled within a figure. For example, label the panels in Figures3 and 4.

All panels are now labeled and font sizes have been increased.

13. In figure 4, what impact does the x-axis outlier (coldest temp) have on the patterns we see? It would be nice to know that the outlier is not driving the trends and to have an explanation of why that accession is much colder than the rest (for example, where is this accession from?).

For the impact of the accessions from the coldest regions (Asian and Swedish accessions) on the climate correlations, see our response to Reviewer #1. Briefly, the results hold if they are excluded from the analysis. We have added geographic information for the accessions coming from the warmest and coldest regions to the caption of Figure 1.

14. The Qst vs Fst analysis was used appropriately but there could be an additional explanation for how to interpret figure 7 in regards to how the gray fields are calculated

We have added more details in the calculation of the neutral expectation in the caption of the figure (now Figure 5), with reference to Materials and methods and the original description of this method in Josephs et al., 2019.

15. Unclear what "high plasticity" means in line 227.

This was removed, see comment #12 from Reviewer #1.

Reviewer #3 (Recommendations for the authors):Line 35: remove so-*

Changed

Line 47-58: the emphasis on CBFs is maybe excessive here as you don't provide compelling evidence for their involvement. Maybe swap for an introduction of the main ecological strategy of Nordic vs Southern adapted accessions?

We have broadened the Introduction with respect to life-history/ecological strategies and also framed the paragraph on cold acclimation thusly.

Line 74-82: Provide the AIC/BIC/deviance or RMSE to justify the use of power vs exponential growth model in materials, but keep it simple here so that the interpretation of the parameter is clearer.

The ANOVA table of the model comparison is now added to Materials and methods.

Figure 2: Upon vernalisattion? I believe you mean cold stratification?

Indeed, thanks for spotting this.

Line 94: This is peripheral but I wonder if there is no biological insight to be gained from fitting a β for each accession. Wouldn't this be informative about the rate of resource accumulation?

The problem with fitting a β for each accession is that β and growth rate (r) describe the speed of rosette size increase together and they are not independent. With that independence I mean that the same size growth pattern can be described by different combinations of r and β. In other words, multiple solutions to the same problem. For this reason we took a pragmatic approach to model r as accession-specific while β is kept fixed. This choice lets r maximally describe the growth-pattern differences between accessions. What, exactly, is physiologically determining β is an interesting question, but outside the scope of this manuscript.

Line 103-128: I understand that some metrics are provided in the figures associated with these sections but it would be clearer to provide some quantitative support (and tests) in the main text.

The correlation coefficients and p-value are now also in the text.

Line 147: how does the metabolite measurement fit?

This refers to the resemblance in geographic pattern, we clarified this in the text.

Line 154: which accessions? Give the rationale for selection based on Figure S4.

The 8 accessions were selected to represent the climatic variation in the full panel. This is clarified in the caption of Figure S8 and in Materials and methods.

Line 162: the null hypothesis of the test is not known.

This is now clarified in Materials and methods.

Line 171-174: I find this argument circular: these genes were identified as differentially expressed in the very same accession I presume (although this is not explicitly stated in the paper).

No, the 251 genes were selected based on published functional studies, so there is no circularity. This has been clarified.

Line 134: relocation of resources towards… I think there could be some information in analysing β's inferred for each accession here.

See response to earlier comment.

Line 360: typo

Fixed.

Line 372: Put the rest of the section in the heritability section.

This section was moved to the description of the heritability calculation, hopefully making it easier to follow (see also comment by Reviewer #1).

Line 392: grammar issue.

Fixed.

Line 462: grammar again.

Fixed.

[Editors’ note: further revisions were suggested prior to acceptance, as described below.]

Reviewer #2 (Recommendations for the authors):The clarity of the manuscript has been improved particularly in the descriptions of the methods and figure design. A few additional analyses to explore subsets of the data that drive trends in growth rate are included which strengthens the overall conclusion that temperature response of vegetative growth is locally adaptive. However, the manuscript has not addressed two of our initial concerns about the broad framing of the manuscript and the gene expression analyses.1. The manuscript has been reframed in the context of ecological and life history strategies with a focus on the growth and survival tradeoff, based on feedback from us and other reviewers. While the growth rate results are potentially consistent with what is expected for a growth/survival tradeoff, there is no evidence for the other side of the tradeoff, survival. The manuscript should make this clear. In addition, the current framing includes references to active and passive plasticity, but we are not convinced that faster growth of accessions from warm temperatures "proves" that slow growth of accessions from cold temperatures is active plasticity (lines 125-127) as opposed to another explanation such as less efficient use of resources in accessions from cold temperatures at a later life stage. Additionally, the manuscript notes that "plant growth is known to slow down with increasing size" when justifying its model choice for estimating growth rates. With this in mind, is it a surprise that the populations with initial larger rosettes now show slower growth?

We agree that we cannot make hard claims on the survival aspect, therefore we were always careful in our wording. We now also added explicitly that we did not measure survival itself. (line 238). At the same time, we note (line 245-246) that reciprocal transplant experiments in *Arabidopsis thaliana* did detect an important contribution of freezing tolerance to fitness variation in sites with strong winters.

We also added a disclaimer paragraph (line 255-259) to make clear that a reduced resource efficiency in later development may cause a similar growth reduction. We still believe that our data is pointing towards active inhibition in trade-off with cold acclimation, however, further work is required to nail down the exact resource relocation and the regulation of this growth inhibition.

2. The gene expression results should be better integrated to support the conclusion that growth rate response is locally adapted. The main results described in the text are (1) the cold acclimation genes are more likely to be differentially expressed between temperatures compared to background genes. This is unsurprising since these genes were initially identified as being differentially expressed in the cold. (2) Gene expression clustering separates accessions from warmer and colder climates and the expression of some clusters is correlated with the temperature of origin. It is unclear if this pattern differs from background genes or the trait divergence expected under neutrality. (3) Genes that were up-regulated in cold (from published experiments) are also generally upregulated in accessions from cold climates. This last result is potentially the clearest link to the adaptive story, but it is not emphasized here and it's also unclear to us if upregulating these genes is a sign of adaptive or maladaptive plasticity. Overall, there is not a clear link between these results and the conclusion about adaptation to cold temperatures.In addition, while Figure 3 —figure supplement 3 does address our concern about pseudoreplication, Figure 3 —figure supplement 2 still reports misleading p values that should not be included due to pseudoreplication.

1) We agree that this is unsurprising, but in our opinion it is important to show that also in this specific setup this group of genes is more affected by temperature than expected by chance.

2) We added figure 3 - figure supplement 5, which is showing that for most clusters (except cluster F in 16ºC), the correlation coefficients are stronger than expected from random sets of background genes. For this we performed 10,000 permutations using random sets of background genes for each of the clusters, in each temperature.

That the correlations are stronger than expected under neutrality is in our opinion proven by the fact that the correlations are (for most clusters) more significant than expected by chance, which was now tested by doing 10.000 permutations of winter temperature correlations for each of the clusters of cold acclimated genes. These p-values are now corrected in figure 3 —figure supplement 2, which is also tackling the problem of the misleading p-values in that figure.

2) These results could be interpreted many ways, however, they do demonstrate that the expression of known cold-response genes differ geographically, consistent with local adaptation. We now note this in the Discussion (line 229-230).

Line Comments:The statements made in lines 41-43 should have citations.

We added citations.

Lines 154-156 seem circular.

We don't think this sentence is circular. The first part of the sentence refers to independent experiments, the second to our own results. We adjusted the sentences to make this more clear.

Line 181 change "both" to "all".

We changed it to 'These', referring to the traits and temperatures discussed in the previous sentence. 'all' would be confusing since there was no adaptive differentiation for all traits in all temperatures.

Line 198-199. The phrase "across the genome we detect adaptive differentiation for certain growth parameters." seems unsupported by the data. While the Qst-Fst results show evidence that phenotypic variation is adaptive, there is still no evidence that this trait is shaped by loci 'across the genome'.

We agree that there is no direct evidence for this and removed the statement.

Line 273 should read 160 µmolm-2s-1?

Indeed, thanks for spotting this.

It is not clear how the growth rate temperature response was calculated. Line 91-92 suggests that it is "the slope between the growth rate at 16ºC and 6ºC" but examining specific points in figure 2B, C, and D makes it look like something else is going on. For example, the far left point has a growth rate of ~0.16 in 16C and ~0.052 in 6C, but the response is ~0.072 when we would expect something closer to 0.1 based on the text.

A value closer to 0.1 is indeed expected when the categorical variables '6C' and '16C' are recoded to 0 and 1. Because we wanted to obtain the slope from 6C towards 16C, we specified temperature to be an ordered factor with 6C the lower order and 16 the higher order. The linear regression used to define the slope works a little bit differently with ordered categorical variables, encoding the categorical variables not to 0 and 1, but to -0.70 and 0.70. The slopes we obtained are thus different but perfectly correlated with the slopes obtained from non-ordered categorical variables. We specified the use of order categorical variables in Materials and methods (line 318-321).

Please include percent variance explained on figures with PCs as axes (Figure 5 and Figure 5 supplement).

The percentages of the genetic variation explained by each of the principal components are added to the respective axis labels.

Figure 1—figure supplement 1 is somewhat confusing with the y axis referring to the experiment timeline on the x-axis and to the inset graph of a single day.

The insert is now put in the upper right corner of the figure, this hopefully will take away the confusion and make clearer that the insert axes stand on their own.

The inclusion of the previously published metabolite data is still somewhat confusing. The results report that metabolite measurements were different between accessions from different climates (line 137) but these results are not presented in a figure anywhere. If the authors want to present previously published data that has been reanalyzed to this paper in a figure, that would be fine. However, without that, it seems like this statement should be moved to the discussion.

We think that mentioning the findings of the previously published metabolite data is important to introduce why we look into the expression of cold acclimation genes. Therefore we prefer to keep this sentence for its introductory value in the gene expression analysis.

We added to the text that the metabolite measurements are presented in a previous publication.